# UI-S1: Advancing GUI Automation via Semi-online Reinforcement Learning

## Abstract

Graphical User Interface (GUI) agents have demonstrated remarkable progress in automating complex user interface interactions through reinforcement learning (RL). However, current approaches face a fundamental dilemma: offline RL enables stable training on pre-collected trajectories, but struggles with multi-step task execution for lack of trajectory-level reward signals; online RL captures these signals through environment interaction, but suffers from sparse rewards and prohibitive deployment costs. To address it, we present **Semi-online Reinforcement Learning**, a novel paradigm that simulates online RL on offline trajectories. During each rollout process, we preserve the original model output within the multi-turn dialogue, where a Patch Module adaptively recovers the divergence between rollout and expert trajectories. To capture long-term training signals, Semi-online RL introduces discounted future returns into the reward computation and optimizes the policy with weighted step-level and episode-level advantages. We further introduce Semi-Online Performance (**SOP**), a metric that aligns better with true online performance, serving as a practical and effective proxy for real-world evaluation. Experiments show that ours **UI-S1-7B** achieves SOTA performance among 7B open-source models across four dynamic benchmarks, with significant gains over the base model (e.g., +12.0% on AndroidWorld, +23.8% on AITW), demonstrating significant progress in bridging the gap between offline training efficiency and online multi-turn reasoning.

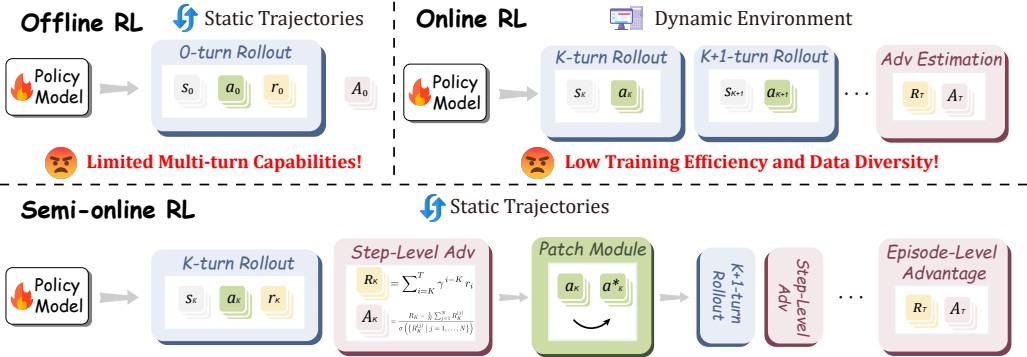

Figure 1: Illustrations of three RL approaches. Our proposed Semi-online RL simulates online RL on offline static trajectories, which enhances multi-turn agent capabilities more efficiently.

## 1 Introduction

Graphical User Interface (GUI) automation represents a critical frontier in developing AI agents that can interact with digital environments as humans do, driven by advances in multimodal large language models that enable complex reasoning and multi-step task execution (Shen et al., 2023; Hu et al., 2025; Zhang et al., 2025a; Wang et al., 2025a; Tang et al., 2025b; Liu et al., 2025a). This evolution has been accelerated by reinforcement learning (RL) techniques that allow agents to

improve through trial-and-error learning, guided by task completion signals (Bai et al., 2024; Lu et al., 2025b; Tang et al., 2025a; Ye et al., 2025; Du et al., 2025; Zheng et al., 2025).

Despite these advances, current reinforcement learning approaches fall into two distinct paradigms (Figure 1), each with critical limitations. Offline RL methods train on pre-collected trajectories with step-wise supervision (Lu et al., 2025b; Luo et al., 2025; Liu et al., 2025b). These approaches leverage large-scale datasets annotated by humans or language models (Li et al., 2024; Lu et al., 2024; Chai et al., 2024), achieving stable training and high single-step accuracy. However, agents trained with offline RL often fail catastrophically when deployed on real-world tasks that require multi-step reasoning and planning. This performance gap arises from two key issues: (1) a mismatch between the offline training and the online evaluation dynamics, particularly regarding whether the original model outputs are consistently recorded into the historical context; and (2) overfitting to local reward signals, leading to ignorance of future or global training objectives.

Online RL methods address this limitation by training agents through direct environment interaction (Lu et al., 2025a; Shi et al., 2025; Ye et al., 2025), learning to handle stochastic transitions with historical context across multiple steps. However, deploying online RL for GUI automation faces prohibitive practical barriers. First, rewards in real-world GUI tasks are typically sparse and delayed, which are often received only at task completion, resulting in inefficient training for complex tasks. Second, enhancing data diversity is inherently difficult: scaling to new environments or tasks requires extensive engineering effort to implement custom verification logic or simulation modules, which can be more labor-intensive than manually curating diverse, high-quality trajectories.

To simultaneously exploit the training efficiency of offline RL, and the long-term optimization target of online RL, we introduce **Semi-online RL**, a novel training paradigm designed for multi-turn interaction learning from pre-collected trajectories. In detail, Semi-online RL preserves original model output including reasoning contexts and historical action within the dialogue state, and then computes step-wise rewards from offline trajectories. Moreover, to improve the comprehensive utilization of trajectory data, a novel Patch Module adaptively recovers the by injecting expert action and synthetic reasoning content. To better capture the current influence on future execution, we further incorporate discounted future reward into step-level advantages and optimize the policy with weighted step-level and episode-level advantages. For efficient multi-turn evaluation, we propose semi-online metric SOP, which demonstrates a stronger correlation with online metrics AndroidWorld ($R^2$=0.934) than traditional offline metrics like AndroidControl-High ($R^2$=0.470) and GUI Odyssey ($R^2$=0.398), as shown in Figure 2 and Figure 9. Experiments demonstrate that ours UI-S1-7B achieves state-of-the-art performance among all open-source 7B models on multi-turn benchmarks, in both dynamic setting (AndroidWorld, AITW, MiniWob++) and static setting (SOP). Notably, UI-S1-7B improves success rates by +12.0 on AndroidWorld and +23.8 on AITW-Gen compared to its base model (i.e., Qwen2.5VL-7B). In addition, it achieves slight gains on out-of-domain single-turn benchmarks (e.g., +1.9 on SS-Pro and +7.1 on GUI Odyssey), validating that Semi-online RL doesn't sacrifice single-turn capabilities.

In summary, our contributions are as follows.

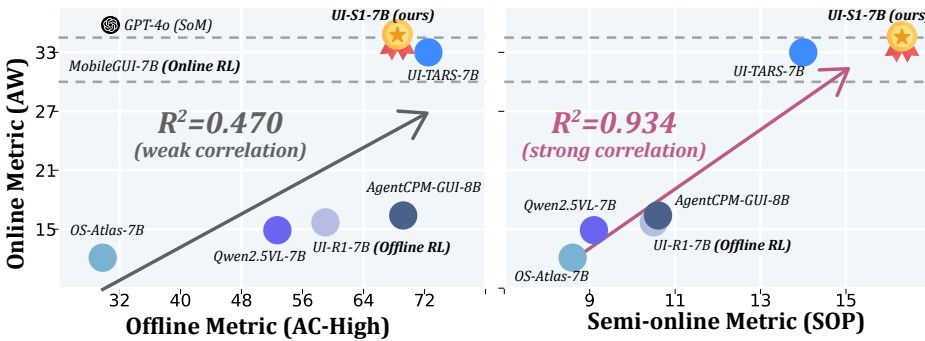

Figure 2: **Left**: Offline metric AC-High demonstrates weak correlation ($R^2$=0.470) with online metric AndroidWorld (AW). **Right**: Our proposed semi-online metric SOP shows stronger correlation ($R^2$=0.934), while ours UI-S1-7B achieves superior performance on both metrics.

- We introduce a training paradigm **Semi-online RL** that simulates online rollout dynamics using static trajectories. A Patch Module is designed to recover from action mismatches by injecting expert actions to maximize trajectory utilization.

- We incorporate discounted future returns and dual-level advantages into policy optimization, which balances step-level accuracy with trajectory-level task completion.

- We propose Semi-Online Performance (SOP), a metric that demonstrates strong correlation with real-world performance. Our model UI-S1-7B achieves state-of-the-art results among 7B models, with +12.0% on AndroidWorld and +23.8% on AITW.

## 2 RELATED WORK

**GUI Agents with Reinforcement Learning** Recent advances in multimodal models have catalyzed significant progress in GUI automation (Hu et al., 2025; Zhang et al., 2025a; Wang et al., 2025a; Tang et al., 2025b; Liu et al., 2025a; Ye et al., 2025). Early approaches rely on supervised fine-tuning with large-scale annotated datasets. AGUVIS (Xu et al., 2024), OS-Atlas (Wu et al., 2024), UGround (Gou et al., 2025), SeeClick (Cheng et al., 2024), and UI-TARS (Qin et al., 2025) leverage millions of annotated GUI elements to achieve impressive single-step accuracy. While these methods demonstrate strong performance on static benchmarks, they suffer from limited generalization to out-of-distribution scenarios and lack the ability to adapt through interaction. Inspired by the success of DeepSeek-R1 (Guo et al., 2025), recent work has begun applying reinforcement learning to GUI automation. UI-R1 (Lu et al., 2025b), GUI-R1 (Luo et al., 2025), and InfiGUI-R1 (Liu et al., 2025b) adopt Group Relative Policy Optimization (GRPO) (Shao et al., 2024) for training, demonstrating improved task completion rates. However, these offline RL methods optimize individual actions independently without maintaining sequential context, leading to poor multi-turn performance in real deployment.

**Multi-Turn Reinforcement Learning** Recognizing the limitations of single-step optimization, recent work has explored multi-turn reinforcement learning through online environment interaction (Feng et al., 2025; Wang et al., 2025b; Dong et al., 2025; Zhang et al., 2025b). ARPO (Lu et al., 2025a) proposes multi-turn policy optimization using GRPO with distributed rollouts and experience replay to handle sparse rewards. The method requires extensive parallel infrastructure and struggles with limited exploration diversity. MobileGUI-RL (Shi et al., 2025) extends GRPO to mobile environments with trajectory-aware advantages and curriculum learning through self-exploration, but faces similar challenges with reward sparsity and deployment costs. Our Semi-online RL addresses these limitations by simulating online dynamics using static trajectories, achieving context continuity without environment access while maintaining training efficiency.

## 3 METHOD

We propose Semi-online RL, a semi-online reinforcement learning framework for training GUI agents that bridges the gap between the stability of offline training and the challenge of online execution. Our approach consists of three key parts. (1) Semi-online rollout (Section 3.2) simulates online interaction dynamics using only offline trajectories; (2) Patch Module (Section 3.3) adaptively recovers the divergence between rollout and expert trajectories; (3) Semi-online Policy Optimization (Section 3.4) optimizes agents through a hierarchical reward structure and dual-level advantages.

### 3.1 PROBLEM FORMULATION

We formulate GUI automation as a multi-turn sequential decision-making problem. Given a high-level instruction $I$ describing the task objective, the agent must interact with the graphical interface to complete the specified goal through a sequence of actions.

At each time step $t$, the agent observes the current state $S_t \in \mathcal{S}$ (typically a screenshot of the interface) and maintains a history of past interactions:

$$H_t = \{(S_1, a_1, \mathcal{T}_1), (S_2, a_2, \mathcal{T}_2), \dots, (S_{t-1}, a_{t-1}, \mathcal{T}_{t-1})\} \tag{1}$$

Figure 3: Illustrations of our proposed Semi-online RL, which consist of semi-online rollout (blue arrows) and dual-level advantage estimation (red arrows).

where $a_i$ represents the executed action and $\mathcal{T}_i$ captures the agent's reasoning process at step $i$. The agent then generates the next action and associated reasoning:

$$a_t, \mathcal{T}_t \sim \pi(\cdot \mid I, S_t, H_t) \tag{2}$$

where $\pi$ denotes the policy model. The environment transitions to the next state according to $S_{t+1} = \mathcal{E}(S_t, a_t)$, and the process continues until task completion or failure.

The fundamental challenge in training GUI agents lies in the mismatch between training and deployment conditions. Traditional offline RL trains on static trajectories where each step conditions on expert demonstrations:

$$H_t^{\text{static}} = \{(S_1^*, a_1^*), \dots, (S_{t-1}^*, a_{t-1}^*)\} \tag{3}$$

In contrast, real-world execution requires the agent to condition on its own generated outputs:

$$H_t^{\text{online}} = \{(S_1, a_1^\pi, \mathcal{T}_1^\pi), \dots, (S_{t-1}, a_{t-1}^\pi, \mathcal{T}_{t-1}^\pi)\} \tag{4}$$

This mismatch causes statically-trained agents to fail catastrophically in multi-turn scenarios, as they never learn to process their own outputs or recover from errors. Online RL addresses this by training with actual environment interaction, but at prohibitive cost. Our Semi-online RL reconciles these approaches by simulating online dynamics using static data.

## 3.2 SEMI-ONLINE ROLLOUT

Given an expert trajectory $\tau^* = \{(S_1^*, a_1^*), \dots, (S_T^*, a_T^*)\}$, we generate training rollouts that maintain policy-generated context while using expert demonstrations for guidance.

During training, we sample $N$ rollouts from the policy model. The $i$-th candidate trajectory is

$$\tau^i = \{(S_1^i, a_1^i), (S_2^i, a_2^i), \dots, (S_T^i, a_T^i)\}, \quad i = 1, \dots, N, \tag{5}$$

The agent maintains its own generated history, serving as subsequent step's condition:

$$H_t^i = \{(S_1^i, a_1^i, \mathcal{T}_1^i), \dots, (S_{t-1}^i, a_{t-1}^i, \mathcal{T}_{t-1}^i)\} \tag{6}$$

At each step, the policy generates action $a_t^i$ based on this self-generated history (from Equation 2). We then use the expert trajectory to approximate environment dynamics:

$$S_{t+1}^i = \begin{cases} S_{t+1}^* & \text{if Matches}(a_t^i, a_t^*) \\ \text{None} & \text{otherwise} \end{cases} \tag{7}$$

When actions match expert demonstrations, we obtain the next state from the expert trajectory and continue with the model's generated history. However, when actions diverge, simple termination would prevent learning from the remaining trajectory steps, particularly resulting in inaccessible later steps which may contain valuable learning signals.

### 3.3 PATCH MODULE FOR TRAJECTORY RECOVERY

To improve the data utilization against early termination, we introduce a Patch Module $\mathcal{P}$ to recover from action mismatches and continue learning from trajectory remainders. When a mismatch occurs at step $t$, the module replaces the incorrect action with the expert action $a_t^*$ and generates synthetic reasoning $\mathcal{T}_t^{\text{patch}}$. The patched components are then integrated into the history, allowing the rollout to continue with $H_{t+1} = H_t \cup \{(S_t, a_t^*, \mathcal{T}_t^{\text{patch}})\}$ (as detailed in Algorithm B). We explore three patching strategies that vary in how synthetic reasoning is generated:

**Thought-Free Patch**  simply injects the expert action without reasoning. This minimal intervention maintains trajectory continuity with an efficient and direct method.

**Off-Policy Thought Patch**  uses an auxiliary model $\mathcal{M}_0$ (e.g., DeepSeek-R1 (Guo et al., 2025)) to generate high-quality reasoning. This ensures coherent thought processes but may introduce distribution shift between the auxiliary and policy models.

**On-Policy Thought Patch**  uses the current policy model $\mathcal{M}$ with expert action hints to generate reasoning. This maintains consistency with the policy's reasoning style while providing correction signals. The prompting strategy for synthetic thought generation is detailed in Appendix I.

### 3.4 SEMI-ONLINE POLICY OPTIMIZATION

Traditional offline RL optimizes only for immediate step-wise accuracy, resulting in multi-turn planning failure. We address this through a hierarchical reward structure and dual-level advantages that capture both immediate and future impacts, inspired by GiGPO (Feng et al., 2025).

For each step in the rollout, we compute a composite reward:

$$r_t = 0.1 \cdot r_{\text{format}} + 0.4 \cdot \mathbb{I}_{[r_{\text{format}}=1]} \cdot r_{\text{type}} + 0.5 \cdot \mathbb{I}_{[r_{\text{format}} \cdot r_{\text{type}}=1]} \cdot r_{\text{acc}} \tag{8}$$

where $r_{\text{format}}$, $r_{\text{type}}$, and $r_{\text{acc}}$ evaluate response formatting, action type correctness, and exact match accuracy respectively.

To capture long-horizon dependencies for multi-turn tasks, we compute discounted future returns:

$$R_t^i = \sum_{k=t}^{t_{\text{end}}} \gamma^{k-t} r_k^i, \quad t_{\text{end}} := \min\left(\max\left\{k \geq t \mid \forall j \in [t,k], \text{ Matches}(a_j^i, a_j^*)\right\} + 1, T\right) \tag{9}$$

where $\gamma \in (0,1)$ weights the influence of future consequences on current decisions, $t_{\text{end}}$ denotes the final step of the natural (w/o patch) trajectory segment where the predicted actions still match the expert, and $T$ is the index of the last step in the full (w/ patch) trajectory.

**Step-Level Advantage**  $A^S(a_t^i)$ captures local optimization signals by comparing returns across trajectories at the same timestep:

$$A^S(a_t^i) = \frac{R_t^i - \mu_t}{\sigma_t} \tag{10}$$

where $\mu_t$ and $\sigma_t$ are computed across all rollouts at step $t$.

**Episode-Level Advantage**  $A^E(\tau^i)$ captures global task completion signals:

$$A^E(\tau^i) = \frac{R(\tau^i) - \mu_\tau}{\sigma_\tau} \tag{11}$$

where $R(\tau^i)$ represents the total trajectory return and is computed as $R_T^i$.

We combine these into a unified advantage that assigns credit at both global and local scales:

$$A(a_t^i) = A^E(\tau^i) + \omega \cdot A^S(a_t^i) \tag{12}$$

Then our Semi-online RL optimizes the policy through the following objective:

$$\mathcal{J}(\theta) = \mathbb{E}_{\substack{\{\tau^i\}_{i=1}^N \sim \mathcal{P}_{\pi_{\theta_{\text{old}}}(\cdot|I)} \\ \{o_{i,t}\}_{t=1}^T \sim \tau^i}} \frac{1}{K} \sum_{i=1}^N \sum_{t=1}^T \sum_{k=1}^{|o_{i,t}|} \min\left(\rho(\theta) A(a_t^i), \text{clip}(\rho(\theta), 1 \pm \epsilon) A(a_t^i)\right) - \beta D_{KL}(\pi_\theta || \pi_{\text{ref}}) \tag{13}$$

where the notation $\overset{\mathcal{P}}{\sim}$ indicates trajectories are generated through our Patch Module-enhanced rollout, $K$ is the total number of tokens, $\rho(\theta) = \frac{\pi_\theta(o_{i,t,k}|I,o_{i,t,<k})}{\pi_{\theta_{\mathrm{old}}}(o_{i,t,k}|I,o_{i,t,<k})}$ is the importance sampling ratio, and $\beta$ controls the KL penalty strength.

To ensure effective learning with sufficient exploration, we enforce minimum advantage variance: $\sigma(\{A(a_t^i)\}) > \eta$, performing dynamic sampling until this diversity threshold is met. In our experiments, we set $\eta = 0.3$.

## 4 EXPERIMENT

### 4.1 EXPERIMENT SETUP

**Baselines.** We compare against three training paradigms using the same dataset: (1) SFT only: supervised fine-tuning on expert demonstrations, (2) Offline RL: traditional offline reinforcement learning with GRPO, conditioning on ground-truth history, and (3) Semi-Online RL only. Our final model combines SFT with Semi-Online RL in a two-stage training pipeline.

**Multi-turn Benchmarks.** To evaluate end-to-end task completion requiring sequential reasoning, we introduce **Semi-Online Performance (SOP)**, an efficient proxy for online evaluation built on AndroidControl-Test (Li et al., 2024). SOP evaluates multi-turn execution by maintaining model-generated history throughout the task. Unlike AndroidControl-High which conditions on ground truth at each step, SOP continues with the model's own outputs, terminating only upon action mismatch. We report Progress (PG) as the average task completion ratio and Task Success Rate (TSR) as the proportion of fully completed tasks (as detailed in Appendix C). To demonstrate GUI agents' real-world performance, we also evaluate on dynamic environments including AndroidWorld (116 tasks) (Rawles et al., 2024), AITW-Gen (300 filtered tasks), AITW-Web (150 filtered tasks) (Bai et al., 2024; Shi et al., 2025), and MiniWob++ (92 tasks) (Liu et al., 2018).

**Single-turn benchmarks** Single-turn evaluates the grounding capability and GUI Understanding capability of the end-to-end GUI model in a single-turn conversation without historical context. We use ScreenSpot-V2 (Cheng et al., 2024) and ScreenSpot-Pro (Li et al., 2025) to evaluate the grounding ability. We also adopt AndroidControl-High (Li et al., 2024) and GUI Odyssey (Lu et al., 2024), for comprehensive GUI understanding evaluation under a high-level instruction. The action type match accuracy (TM), grounding accuracy rate (GR) and step success rate (SR) are reported.

Table 1: Results on Multi-turn Benchmarks. * denotes the result using prompt in Appendix H.

| | SOP | | AITW-Gen | AITW-Web | MiniWob++ | AW |
|---|---|---|---|---|---|---|
| | PG | TSR | | | | |
| *Closed-source Models* | | | | | | |
| Gemini-Pro-1.5 (SoM) (Team et al., 2024) | – | – | – | – | – | 22.8 |
| Claude Computer Use (Anthropic, 2024) | – | – | – | – | – | 27.9 |
| GPT-4o (SoM) (Hurst et al., 2024) | – | – | – | – | – | **34.5** |
| *Open-source 7B/8B Models* | | | | | | |
| OS-Genesis-7B (Sun et al., 2024) | 7.6 | 3.0 | 14.5 | 7.8 | 19.8 | 17.4 |
| OS-Atlas-7B (Wu et al., 2024) | 14.3 | 8.6 | 45.6 | 17.9 | 35.2 | 12.1 |
| Qwen2.5VL-7B (Bai et al., 2025) | 17.4 | 9.8 | 49.0 | 20.0 | 54.0 | 22.0 |
| AgentCPM-GUI-8B (Zhang et al., 2025c) | 17.1 | 10.6 | 58.6 | 15.2 | 37.8 | 16.4 |
| MobileGUI-7B (Shi et al., 2025) | – | – | 65.3 | 22.7 | – | 30.0 |
| UI-TARS-7B (Qin et al., 2025) | 28.1 | 14.0 | 64.9 | 28.1 | 58.7 | 33.0 |
| *Open-source 32B/72B Models* | | | | | | |
| Qwen2.5VL-32B (Bai et al., 2025) | 17.8 | 10.2 | 42.7 | 24.7 | **70.1** | 31.5 |
| Aguvis-72B Xu et al. (2024) | – | – | – | – | 66.0 | 26.1 |
| *Ours 7B Models* | | | | | | |
| Qwen2.5VL-7B (Base)* | 16.8 | 9.1 | 50.5 | 28.8 | 54.0 | 14.9 |
| w/ SFT | 17.0 | 9.3 | 58.9 | 28.5 | 46.7 | 21.7 |
| w/ Offline RL | 18.3 | 10.5 | 54.6 | 19.8 | 53.3 | 15.7 |
| w/ Semi-online RL only | 30.6 | 16.0 | 70.2 | 36.3 | 57.6 | 30.4 |
| UI-S1-7B | **32.4** | **16.3** | **74.3** | **40.2** | 60.9 | 34.0 |
| Δ (vs Base) | *+15.6* | *+7.2* | *+23.8* | *+11.4* | *+6.9* | *+19.1* |

Table 2: Results on single-turn benchmarks.

| | SS-V2 | SS-Pro | AC-High | | | GUI Odyssey | | |
| --- | --- | --- | --- | --- | --- | --- | --- | --- |
| | | | TM | GR | SR | TM | GR | SR |
| *Closed-source Models* | | | | | | | | |
| GPT-4o (Hurst et al., 2024) | 18.3 | 0.8 | 66.3 | 0.0 | 20.8 | 34.3 | 0.0 | 3.3 |
| Claude-computer-use (Anthropic, 2024) | 83.0 | 17.1 | 63.7 | 0.0 | 12.5 | 60.9 | 0.0 | 3.1 |
| SeeClick (Cheng et al., 2024) | 55.1 | 1.1 | 82.9 | 62.9 | 59.1 | 71.0 | 52.4 | 53.9 |
| *Open-source Models* | | | | | | | | |
| OS-Atlas-4B (Wu et al., 2024) | 71.9 | 3.7 | 49.0 | 49.5 | 22.8 | 49.6 | 34.6 | 20.3 |
| Qwen2.5VL-3B (Bai et al., 2025) | 80.9 | 28.7 | 47.8 | 46.5 | 38.9 | 37.4 | 26.5 | 26.7 |
| UI-R1-3B (Lu et al., 2025b) | 85.4 | 17.8 | 57.9 | 55.7 | 45.4 | 52.2 | 34.5 | 32.5 |
| GUI-R1-3B (Luo et al., 2025) | 85.0 | 28.6 | 58.0 | 56.2 | 46.6 | 54.8 | 41.5 | 41.3 |
| OS-Genesis-7B (Sun et al., 2024) | – | – | 65.9 | – | 44.4 | 11.7 | – | 3.6 |
| OS-Atlas-7B (Wu et al., 2024) | 84.1 | 18.9 | 57.4 | 54.9 | 29.8 | 60.4 | 39.7 | 27.0 |
| Aguvis-7B (Xu et al., 2024) | 81.8 | 22.9 | 65.6 | – | 54.2 | 26.7 | – | 13.5 |
| GUI-R1-7B (Luo et al., 2025) | 88.2 | 31.3 | 71.6 | 65.6 | 51.7 | 65.5 | 43.6 | 38.8 |
| AgentCPM-GUI-8B (Zhang et al., 2025c) | – | – | 77.7 | – | 69.2 | 90.8 | – | 75.0 |
| UI-TARS-7B (Qin et al., 2025) | 91.6 | 35.7 | 83.7 | 80.5 | 72.5 | 94.6 | 90.1 | 87.0 |
| *Ours 7B Models* | | | | | | | | |
| Qwen2.5VL-7B (Base) | 89.0 | 28.7 | 62.2 | 72.5 | 52.7 | 67.4 | 56.3 | 52.4 |
|   w/ SFT | 90.1 | 29.6 | 66.8 | 74.3 | 56.1 | 56.9 | 61.5 | 43.2 |
|   w/ Offline RL | 88.4 | 29.2 | 69.7 | 68.2 | 59.0 | 62.5 | 50.2 | 48.7 |
|   w/ Semi-online RL only | 89.7 | 30.2 | 77.6 | 71.3 | 66.8 | 74.5 | 58.9 | 56.3 |
| UI-S1-7B | 90.1 | 30.6 | 79.9 | 73.4 | 68.2 | 76.3 | 61.7 | 59.5 |
|   $\Delta$ (vs Base) | *+1.1* | *+1.9* | *+17.7* | *+0.9* | *+15.5* | *+8.9* | *+5.4* | *+7.1* |

## 4.2 MAIN RESULTS

**Multi-turn Performance.** As shown in Table 1, UI-S1-7B establishes a new state-of-the-art among 7B/8B open-source models across all evaluated multi-turn benchmarks. Compared to Qwen2.5VL-7B, UI-S1-7B achieved substantial improvements: +19.1% on AndroidWorld and +23.8% on AITW-Gen. Remarkably, our UI-S1-7B outperforms strong baselines such as MobileGUI-7B and also delivers competitive results on AndroidWorld (34.0%) compared with significantly larger open-source models like Qwen2.5VL-32B (31.5%) and Aguvis-72B (26.1%), as well as closed-source models such as GPT-4o (34.5%). Despite its enhanced planning and reasoning for UI navigation, UI-S1-7B exhibits limitations in tasks requiring precise numerical computation and abstract logical reasoning, as detailed in Table 10.

The comparison between training paradigms reveals critical insights. While SFT improves over the base model, it shows slight gains on dynamic benchmarks (21.7% on AW). Traditional Offline RL actually degrades model performance (53.3% on MiniWob++) compared to the base model, demonstrating its limited capabilities on real-world generalization. Our approach (Semi-Online RL only) achieves 30.4% on AW, and Semi-Online RL (w/ SFT) reaches 34.0%, validating its generalization.

**Single-turn Performance.** Table 2 shows that Semi-Online RL maintains competitive single-turn performance while excelling at multi-turn tasks. Our model achieves consistent improvements over the base: +15.5% on AndroidControl-High SR and +7.1% on GUI Odyssey SR. However, offline RL models like AgentCPM-GUI-8B excel at single-turn tasks but struggle with multi-turn execution (16.4 on AW). This demonstrates that Semi-Online RL successfully bridges both capabilities rather than trading one for the other.

## 4.3 ANALYSIS OF PATCH MODULE STRATEGIES

We present the results of patch strategies across different data scales and thresholds in Table 3.

**Impact of Patch Threshold.** The patch threshold $\epsilon$ controls how many mismatches are recovered before termination. Results demonstrate that increasing $\epsilon$ consistently improves both SOP and AndroidWorld metrics. With 1000 training samples, SOP-Score increases from 22.3 ($\epsilon$=0) to 25.7

Table 3: Performance comparison for different $\epsilon$ values with varying data sizes (200, 500, 1000 from left to right). Each table shows results for SOP and AW under three patching strategies.

| $\epsilon$ | SOP | | | AW | $\epsilon$ | SOP | | | AW | $\epsilon$ | SOP | | | AW |
|---|---|---|---|---|---|---|---|---|---|---|---|---|---|---|
| | PG | TSR | Score | | | PG | TSR | Score | | | PG | TSR | Score | |
| *Thought-Free Patch* | | | | | *Thought-Free* | | | | | *Thought-Free Patch* | | | | |
| 0 | 26.3 | 14.3 | 20.3 | 21.0 | 0 | 28.0 | 14.8 | 21.4 | 27.2 | 0 | 29.6 | 15.0 | 22.3 | 30.0 |
| 1 | 27.9 | 15.1 | 21.5 | 24.0 | 1 | 28.5 | 15.7 | 22.1 | 29.1 | 1 | 32.4 | 16.3 | 24.4 | 34.0 |
| 2 | 29.1 | 16.5 | 22.8 | 25.4 | 2 | 31.6 | 16.5 | 24.1 | 31.5 | 2 | 32.6 | 16.8 | 24.7 | 33.9 |
| $\infty$ | 30.4 | 16.7 | 23.6 | 25.6 | $\infty$ | 33.8 | 17.0 | 25.4 | 30.8 | $\infty$ | 34.4 | 17.0 | 25.7 | 34.5 |
| *Off-Policy Thought Patch* | | | | | *Off-Policy Thought Patch* | | | | | *Off-Policy Thought Patch* | | | | |
| 0 | 26.3 | 14.3 | 20.3 | 21.0 | 0 | 28.0 | 14.8 | 21.4 | 27.2 | 0 | 29.6 | 15.0 | 22.3 | 30.0 |
| 1 | 24.0 | 12.9 | 18.5 | 19.7 | 1 | 28.5 | 12.5 | 20.5 | 25.0 | 1 | 29.5 | 12.0 | 20.8 | 24.6 |
| 2 | 28.1 | 14.9 | 21.5 | 25.0 | 2 | 30.0 | 13.5 | 21.8 | 26.0 | 2 | 31.6 | 12.6 | 22.1 | 25.3 |
| $\infty$ | 30.2 | 13.3 | 21.8 | 24.0 | $\infty$ | 30.5 | 14.0 | 22.3 | 24.0 | $\infty$ | 31.8 | 13.3 | 22.6 | 24.0 |
| *On-Policy Thought Patch* | | | | | *On-Policy Thought Patch* | | | | | *On-Policy Thought Patch* | | | | |
| 0 | 26.3 | 14.3 | 20.3 | 21.0 | 0 | 28.0 | 14.8 | 21.4 | 27.2 | 0 | 29.6 | 15.0 | 22.3 | 30.0 |
| 1 | 28.7 | 15.3 | 22.0 | 25.0 | 1 | 31.0 | 15.2 | 23.1 | 28.2 | 1 | 32.9 | 16.7 | 24.8 | 31.4 |
| 2 | 29.4 | 16.0 | 22.7 | 24.9 | 2 | 32.0 | 16.7 | 24.4 | 29.8 | 2 | 33.1 | 17.4 | 25.3 | 31.9 |
| $\infty$ | 30.3 | 17.1 | 23.7 | 26.9 | $\infty$ | 33.2 | 17.2 | 25.2 | 31.5 | $\infty$ | 34.4 | 17.8 | 26.1 | 32.8 |

Figure 4: Data scaling for different $\epsilon$ for Thought-free patch, with SOP-score reported.

Figure 5: Actor entropy during training process with different patch method and threshold.

Figure 6: Comparison of Semi-online RL (with different $\gamma$) and Offline RL during training.

($\epsilon=\infty$) for Thought-Free Patch, representing a 15% relative improvement. This gain stems from increased exposure to later trajectory steps, as higher $\epsilon$ values enable learning from previously inaccessible trajectory segments. Figure 5 reveals that larger $\epsilon$ values maintain greater policy entropy during training, indicating more diverse exploration and preventing premature convergence. We select $\epsilon=1$ as optimal, achieving 34.0% on AndroidWorld while minimizing computational overhead.

**Comparison of Patch Methods.** Three patching strategies exhibit distinct trade-offs between performance and efficiency (from Figure 10). On-Policy Thought Patch achieves the highest SOP scores (26.1 at $\epsilon=\infty$) by maintaining reasoning consistency with the policy model. Thought-Free Patch delivers competitive performance (25.7) with significantly lower computational cost, requiring no additional inference for synthetic reasoning generation. Off-Policy Thought Patch underperforms (22.6) due to distribution mismatch between the auxiliary model's reasoning style and the policy model's expectations. Based on these results and efficiency considerations, we adopt Thought-Free Patch with $\epsilon=1$ for our final configuration.

## 4.4 ANALYSIS OF TRAINING DYNAMICS

**Scaling Law Performance.** Figure 4 reveals the data scaling performance of Semi-Online RL across different patch configurations. The performance follows an exponential scaling law $y = A + B \cdot e^{C+kx}$, where the scaling coefficient $k$ increases with $\epsilon$ from $-1.13$ to $-0.73$. This indicates that larger $\epsilon$ values not only improve absolute performance but also enhance data efficiency, enabling more effective learning from each training sample. The improved scaling stems from better utilization of trajectory data, as the Patch Module enables learning from steps that would otherwise be terminated after action mismatches.

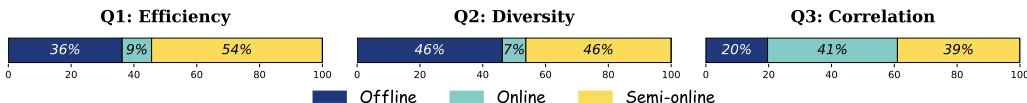

Figure 7: Comparison of offline (AC-High), online (AndroidWorld), and semi-online (SOP) methods across three dimensions: efficiency (inverse rollout time cost), diversity (number of tasks), and correlation (vs online performance).

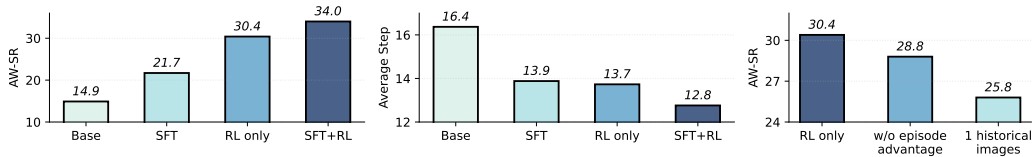

Figure 8: **Left**: Performance of different training paradigm combinations. **Middle**: Average steps to complete AndroidWorld tasks. **Right**: Ablations on episode advantages and historical images.

**Semi-Online Performance Metric.** Figure 7 validates SOP as an effective proxy for real-world evaluation. We compare three evaluation paradigms across efficiency (inverse time cost), diversity (number of tasks), and correlation with online performance. SOP achieves the highest correlation with AndroidWorld ($R^2$=0.934), substantially outperforming AndroidControl-High ($R^2$=0.470) while requiring minimal evaluation time. This strong correlation confirms that maintaining model-generated history during evaluation accurately captures the multi-turn dynamics. The metric fills a critical gap between fast but unrealistic offline evaluation and accurate but expensive online testing.

### 4.5 ABLATION STUDIES

**Discount Factor Analysis.** The results in Figure 6 demonstrate the importance of future reward discounting in Semi-Online RL. Our approach increases the task success rate during training steps while traditional Offline RL exhibits opposite behavior. This divergence highlights a fundamental difference: Semi-Online RL's historical context continuity enables effective multi-turn paradigms learning, while Offline RL ignores long-horizon training signals. Among different $\gamma$ in our setting, performance peaks at $\gamma$=0.5. Setting $\gamma$=0 (no future rewards) yields the worst results, confirming that long-horizon optimization is essential for multi-turn tasks.

**Training Paradigm.** We also conduct ablation studies on training paradigms in Figure 8. Combining SFT with Semi-Online RL outperforms either method alone, achieving 34.0% on AndroidWorld compared to 30.4% for Semi-Online RL only and 21.7% for SFT only. The combined approach also reduces average task completion steps (middle panel), eliminating redundant actions with better planning. Additional ablations (right panel) confirm that both episode-level advantages and maintaining multiple historical images contribute to performance, validating our training setup. More ablations about the hyper-parameter and the reward design are shown in Appendix E. We also conduct experiments on 3B and 32B models to investigate the effect of model scale and demonstrate the generalization capability of our method (as shown in Table 6). Detailed task analysis and case study are shown in Appendix G.

## 5 CONCLUSION

In this work, we present Semi-online Reinforcement Learning (Semi-online RL), a novel training paradigm that bridges the advantages of offline and online reinforcement learning for GUI automation agents, enabling stable yet long-horizon-capable policy optimization. Experimental evaluation shows that our UI-S1-7B achieves state-of-the-art results among open-source 7B-scale models, with substantial improvements across both dynamic and static multi-turn benchmarks, without compromising single-turn performance. Our findings highlight the promise of Semi-online RL as an effective and scalable training framework for real-world GUI agents.

## ETHICS STATEMENT

We have conducted our research in full accordance with the ICLR Code of Ethics. Our primary ethical focus was ensuring the privacy and integrity of the data used. All GUI datasets employed in our SFT and RL training stages are sourced exclusively from public, open-source benchmarks. These datasets are human-collected and specifically designed for research, and we have verified that they are free from any user privacy, personally identifiable information (PII), or confidential commercial data. While our work aims to advance positive applications like GUI automation, we acknowledge the potential for dual-use of this technology and encourage the community to use our open-source contributions responsibly. We are committed to transparency and the ethical development of AI.

## REPRODUCIBILITY STATEMENT

To ensure full reproducibility of our work, we provide comprehensive details on our models, data, and experimental setup. Our method is built upon the open-source Qwen2.5-VL model family, and we validate its effectiveness by fine-tuning models of 3B, 7B, and 32B parameters. The training process, including the SFT and RL stages, utilizes approximately 2,000 samples sourced from public, unmodified, open-source benchmarks. All experiments were conducted on a cluster of 32 A100 GPUs. The complete source code, including detailed setup instructions, training scripts, and documentation, is available in the supplementary materials to facilitate replication.

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

# A NOTATION DEFINITION

Table 4: Notation Definition in Section 3.

| Symbol | Description |
|---|---|
| $\mathcal{M}_0$ | Assist model used for thought patching |
| $\mathcal{M}$ | Policy model |
| *prompt* | Prompt template for thought generation (as shown in Appendix I |
| $a_t$ | Predicted action at step $t$ |
| $a_t^*$ | Expert action at step $t$ |
| $\mathcal{T}_t$ | Thought representation at step $t$ |
| $\mathcal{F}$ | Patch function that outputs a (possibly corrected) action and thought |
| $I$ | High-level GUI instruction |
| $S_t$ | Current observation (e.g., screenshot) at step $t$ |
| $H_t$ | Full history up to step $t$ including $(S, a, \mathcal{T})$ tuples |
| $r_{\text{format}}$ | Binary score (0 or 1) for correct output format |
| $r_{\text{type}}$ | Binary score (0 or 1) for correct predicted action type |
| $r_{\text{acc}}$ | Binary score (0 or 1) for exact action match with ground truth |
| $r_t$ | Step-wise reward at time $t$ |
| $\mathbb{I}_{[\cdot]}$ | Indicator function that equals 1 only if condition is true |
| $\gamma$ | Discount factor for return computation ($0 < \gamma < 1$) |
| $R_t^i$ | Discounted return of $i$-th trajectory starting from step $t$ |
| $N$ | Number of trajectories sampled in a batch |
| $A^S(\boldsymbol{a}_t^i)$ | Step-level advantage for action $\boldsymbol{a}_t^i$ |
| $A^E(\boldsymbol{\tau}^i)$ | Episode-level advantage for trajectory $\tau^i$ |
| $R(\tau^{(j)})$ | Episode return of trajectory $j$ |
| $t_{\text{end}}$ | Last step of a natural trajectory segment |
| $T$ | Last step index of a trajectory |
| $\sigma(\cdot)$ | Standard deviation function |
| $\omega$ | Weight balancing episode- and step-level advantages |
| $A(\boldsymbol{a}_t^i)$ | Combined group-in-group advantage |
| $K$ | Total number of tokens in the current batch |
| $o_{i,t}$ | Model output sequence (tokens) at step $t$ of trajectory $i$ |
| $o_{i,t,k}$ | $k$-th token of $o_{i,t}$ |
| $q$ | Conditioning input (e.g., prompt including state/action history) |
| $\rho(\theta)$ | Importance sampling ratio between new and old policies |
| $\theta$ | Current policy parameters |
| $\theta_{\text{old}}$ | Policy parameters before update (rollout policy) |
| $\pi_{\text{ref}}$ | Reference policy for KL regularization |
| $\beta$ | Coefficient for KL divergence penalty |
| $\eta$ | Minimum standard deviation threshold for advantage diversity |

# B  PATCH MODULE

---

**Algorithm 1** Semi-Online Rollout with **Patch Module**

---

**Input:**
   $\pi_{\theta_{\text{old}}}$ : initial policy model
   $\tau^* = \{(S_1^*, a_1^*), \dots, (S_T^*, a_T^*)\}$ : offline trajectory
**Output:** $\tau = \{(S_1, a_1), (S_2, a_2), \dots\}$ : trajectory rollout
Initialize $H_1 \leftarrow \varnothing, \tau \leftarrow \varnothing, c \leftarrow 0$
$S_1 \leftarrow S_1^*$
**for** $t = 1$ **to** $T$ **do**
   $a_t, \mathcal{T}_t \sim \pi_{\theta_{\text{old}}}(\cdot \mid S_t, H_t)$           ▷ Sample output from Equation 2
   $a_t^*, S_{t+1}^* \sim \tau^*$           ▷ Fetch ground truth

      Patch Module:
   **if** $a_t = a_t^*$ **then**
      $(a_t^{\text{patch}}, \mathcal{T}_t^{\text{patch}}) \leftarrow a_t, \mathcal{T}_t$           ▷ Continue rollout (no patching needed)
   **else if** $c < \epsilon$ **then**
      $a_t^{\text{patch}}, \mathcal{T}_t^{\text{patch}} \leftarrow \mathcal{F}(a_t, \mathcal{T}_t)$           ▷ Apply patch function defind in Table 5
      $c \leftarrow c + 1$
   **else**
      $\tau \leftarrow \tau \cup (S_t, a_t)$
      $a_t^{\text{patch}}, \mathcal{T}_t^{\text{patch}}, S_{t+1} \leftarrow$ NONE           ▷ Terminate rollout due to max patches reached
      **break**
   **end if**

   **if** $S_{t+1} =$ NONE **then**
      **break**
   **end if**
   $S_{t+1} \leftarrow S_{t+1}^*$
   $H_{t+1} \leftarrow H_t \cup \{(S_t, a_t^{\text{patch}}, \mathcal{T}_t^{\text{patch}})\}$
   $\tau \leftarrow \tau \cup (S_t, a_t^{\text{patch}})$
   $H_t \leftarrow H_{t+1}, S_t \leftarrow S_{t+1}$           ▷ Prepare for next step
**end for**
**Output:** $\tau$

---

Table 5: Different thought patch methods. $\mathcal{M}_0$ denotes the auxiliary model and $\mathcal{M}$ denotes the policy model.

| Patch Method | Function Definition |
|---|---|
| Thought-Free Patch | $\mathcal{F}(a_t, \mathcal{T}_t) = (a_t^*, \emptyset)$ |
| Off-Policy Thought Patch | $\mathcal{F}(a_t, \mathcal{T}_t) = (a_t^*, \mathcal{M}_0(I, a_t^*, S_t))$ |
| On-Policy Thought Patch | $\mathcal{F}(a_t, \mathcal{T}_t) = (a_t^*, \mathcal{M}(I, a_t^*, H_t, S_t))$ |

## C    SOP

**Definition**   Let $N$ be the total number of tasks. For the $i$-th task, let $s_i$ denote the number of successful steps, and $t_i$ denote the total number of steps in its expert trajectory. We define the following metrics: $\text{PG} = \frac{1}{N} \sum_{i=1}^{N} \frac{s_i}{t_i}$, $\text{TSR} = \frac{1}{N} \sum_{i=1}^{N} \mathbb{I}[s_i = t_i]$, and $\text{Score} = \frac{\text{PG}+\text{TSR}}{2}$. Here, $\mathbb{I}[\cdot]$ is the indicator function, which equals $1$ if the condition inside the brackets is true and $0$ otherwise.

**SOP's alignment with online metrics**   We also compare other online metrics and offline metrics GUI Odyssey with SOP in Figure 9, which demonstrates SOP's strong correlation with online metrics.

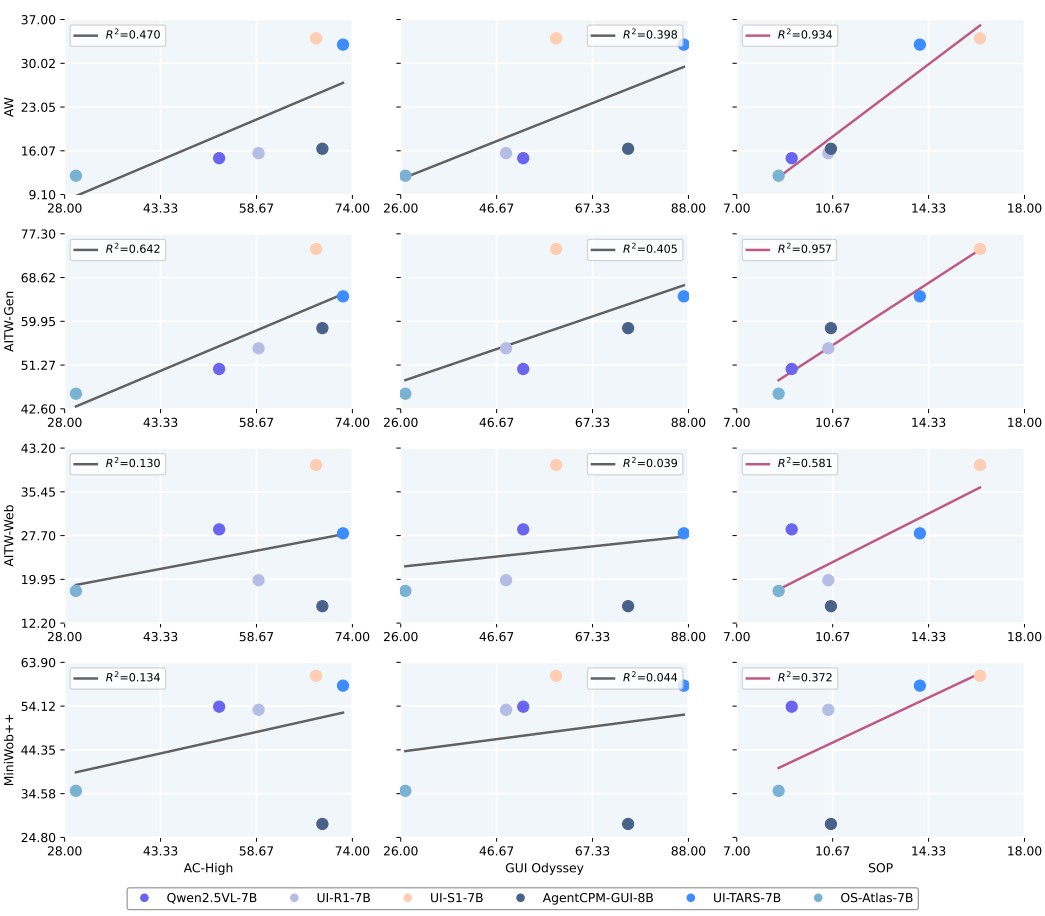

Figure 9: Overall comparisons of online metrics (AW, AITW-Gen, AITW-Web, MiniWob++) with offline metrics (AC-High, GUI Odyssey) and semi-online metric (SOP). **Left**: AC-High demonstrates weak correlation with online metrics. **Middle**: GUI Odyssey demonstrates weak correlation with online metrics. **Right**: Ours SOP demonstrates stronger correlation with online metrics. proposed SOP shows stronger correlation ($R^2$=0.934).

For the linear regression analyses in Figure 2 and Figure 9, the coefficient of determination, denoted as $R^2$, is defined as $R^2 = 1 - \frac{\text{SS}_{\text{res}}}{\text{SS}_{\text{tot}}}$, where $\text{SS}_{\text{res}}$ (Residual Sum of Squares) is $\text{SS}_{\text{res}} = \sum_{i=1}^{n} (y_i - \hat{y}_i)^2$, and $\text{SS}_{\text{tot}}$ (Total Sum of Squares) is $\text{SS}_{\text{tot}} = \sum_{i=1}^{n} (y_i - \bar{y})^2$. Here, $n$ is the number of observations; $y_i$ is the observed value of the dependent variable for the $i$-th data point; $\hat{y}_i$ is the corresponding predicted value from the regression model; and $\bar{y}$ is the mean of all observed values. The $R^2$ metric ranges from 0 to 1 and represents the proportion of variance in the dependent variable explained by the independent variable(s)—higher values indicate a better fit.

# D    MODEL SIZE SCALING

To assess the impact of model scale, we evaluate our method on Qwen2.5-VL-3B and Qwen2.5-VL-32B. As shown in Table 6, Semi-online RL yields consistent performance gains across both model sizes, demonstrating its strong generalizability. Notably, the margin of improvement diminishes as the model scale increases.

Table 6: Performance comparison on different model sizes (3B, 7B, 32B) w/o SFT cold start.

| Model | $SOP_{PG}$ | $SOP_{TSR}$ | $SOP_{Score}$ | $AW_{SR}$ | AVG |
|---|---|---|---|---|---|
| QwenVL-2.5-3B | 3.4 | 1.4 | 2.4 | 5.0 | 3.7 |
| UI-S1-3B | $14.7_{332\%\uparrow}$ | $6.5_{364\%\uparrow}$ | $10.6_{342\%\uparrow}$ | $13.1_{162\%\uparrow}$ | $11.9_{222\%\uparrow}$ |
| QwenVL-2.5-7B | 16.8 | 9.1 | 13.0 | 14.9 | 14.0 |
| UI-S1-7B | $30.6_{82\%\uparrow}$ | $16.0_{76\%\uparrow}$ | $23.3_{79\%\uparrow}$ | $30.4_{104\%\uparrow}$ | $26.9_{92\%\uparrow}$ |
| QwenVL-2.5-32B | 17.8 | 10.2 | 14.0 | 28.3 | 21.2 |
| UI-S1-32B | $35.9_{102\%\uparrow}$ | $18.9_{85\%\uparrow}$ | $27.4_{96\%\uparrow}$ | $38.9_{37\%\uparrow}$ | $33.2_{57\%\uparrow}$ |

# E    OTHER ABLATIONS

**Hyper-parameter**    We conduct an ablation study to determine the optimal values for key hyper-parameters. As detailed in Table 7, we explore different settings for $\gamma \in \{0, 0.5\}$, $\omega \in \{0, 0.5, 1\}$, and $\eta \in \{0.1, 0.3, 0.5\}$. Based on the empirical results, we adopt $\gamma = 0.5$, $\omega = 1$, and $\eta = 0.3$ for the final training configuration.

Table 7: Ablation on $\gamma$ (future reward discount), $\omega$ (advantage weight), $\eta$ (DAPO threshold) with $\epsilon$ (patch threshold) fixed at 0, data size as 1000 and training epoch as 1. SOP is reported.

| $\gamma$ | $\omega$ | $\eta$ | $SOP_{PG}$ | $SOP_{TSR}$ | $\gamma$ | $\omega$ | $\eta$ | $SOP_{PG}$ | $SOP_{TSR}$ |
|---|---|---|---|---|---|---|---|---|---|
| 0.0 | 0.0 | 0.1 | 22.2 | 11.0 | 0.5 | 0.0 | 0.1 | 26.8 | 13.8 |
| 0.0 | 0.0 | 0.3 | 22.3 | 10.8 | 0.5 | 0.0 | 0.3 | 26.1 | 14.6 |
| 0.0 | 0.0 | 0.5 | 21.7 | 11.4 | 0.5 | 0.0 | 0.5 | 27.0 | 13.9 |
| 0.0 | 0.5 | 0.1 | 22.7 | 12.2 | 0.5 | 0.5 | 0.1 | 26.9 | 14.2 |
| 0.0 | 0.5 | 0.3 | 23.3 | 12.5 | 0.5 | 0.5 | 0.3 | 27.3 | 14.0 |
| 0.0 | 0.5 | 0.5 | 22.5 | 10.2 | 0.5 | 0.5 | 0.5 | 27.5 | 14.5 |
| 0.0 | 1.0 | 0.1 | 20.6 | 11.8 | 0.5 | 1.0 | 0.1 | 26.5 | 14.8 |
| 0.0 | 1.0 | 0.3 | 22.2 | 11.2 | 0.5 | 1.0 | 0.3 | 27.9 | 15.4 |
| 0.0 | 1.0 | 0.5 | 22.8 | 12.1 | 0.5 | 1.0 | 0.5 | 28.4 | 14.5 |

**Future reward**    We conduct an ablation study to investigate the optimal choice for $t_{\text{end}}$, as defined in Equation 9. As presented in Table 8, the results demonstrate that setting $t_{\text{end}}$ to the final step of a natural trajectory segment yields superior performance compared to using the end of the entire trajectory.

Table 8: Ablation on $t_{\text{end}}$ with AndroidWorld success rate reported.

| $t_{\text{end}}$ | $\epsilon = 0$ | $\epsilon = 1$ | $\epsilon = 2$ | $\epsilon = \infty$ |
|---|---|---|---|---|
| $T$ | 25.6 | 27.9 | 27.7 | 27.4 |
| $\min(\max\{k \geq t \mid \forall j \in [t, k], \text{Matches}(a_j^i, a_j^*)\} + 1, T)$ | 25.6 | 28.0 | 28.9 | 28.4 |

# F    TRAINING DETAILS

**Training parameters**    Our UI-S1-7B is first Supervised Fine-Tuned (SFT) on Qwen2.5VL-7B, trained on data from AndroidControl-Train (Li et al., 2024) and Amex (Chai et al., 2024), then optimized using Semi-online RL with the thought-free patch mechanism. The training parameters are listed in Table 9.

Table 9: Key Training Hyper-parameters

| Parameter | Value |
|---|---|
| train_batch_size | 32 |
| max_prompt_length | 12288 |
| $\gamma$ (future reward discount) | 0.5 |
| $\omega$ (advantage weight) | 1.0 |
| $\epsilon$ (patch threshold) | 1 |
| $\eta$ (DAPO threshold) | 0.3 |
| historical images | 2 |
| learning rate | $1 \times 10^{-6}$ |
| ppo_mini_batch_size | 32 |
| fixed_num_mini_batches | 4 |
| ppo_micro_batch_size_per_gpu | 1 |
| kl_loss_coef | $1 \times 10^{-4}$ |
| n_gpus_per_node | 8 |
| nnodes | 4 |
| total_epochs | 5 |

**Training hours**    We analyze the training overhead of different patch methods, measured in GPU hours, as depicted in Figure 10. Although the on-policy method yields a slight improvement in SOP performance, it incurs a significant 2.3-fold increase in training time compared to other approaches. To strike a balance between effectiveness and efficiency, we therefore select the thought-free patch method for our final model.

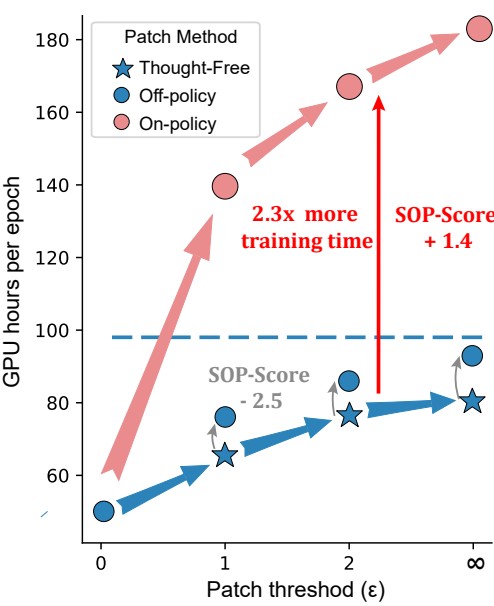

Figure 10: Training GPU hours of different patch methods and patch threshold.

# G CASE STUDY

## G.1 TASK ANALYSIS

As detailed in Table 10, our UI-S1-7B model demonstrates substantial performance gains over the Qwen2.5VL-7B baseline across a majority of task categories. The improvements are particularly pronounced in tasks requiring multi-step interactions and complex comprehension, such as multi_app (+1.00), search (+0.73 on Easy), requires_setup (+0.67 on Easy), and complex_ui_understanding. These results strongly suggest its enhanced planning and reasoning capabilities for navigating complex user interfaces. Nevertheless, challenges persist in domains that demand specialized skills. For instance, both models struggle with game_playing and math_counting tasks. We attribute this to the inherent limitations of small-scale vision-language models in handling precise numerical computation and abstract logical reasoning. We also showcase a failure case of math_counting in Section G.4. In this case, while UI-S1-7B was able to remember the numbers it encountered, it made an error at step 11, calculating 9*10*9*5*5 as 2250.

| Tags | Qwen2.5VL-7B | | | UI-S1-7B | | |
|---|---|---|---|---|---|---|
| | **Easy** | **Medium** | **Hard** | **Easy** | **Medium** | **Hard** |
| complex_ui_understanding | 0.00 | 0.00 | 0.00 | $0.17_{+0.17}$ | $0.20_{+0.20}$ | $0.14_{+0.14}$ |
| data_edit | 0.09 | 0.00 | 0.00 | $0.64_{+0.55}$ | $0.14_{+0.14}$ | $0.00_{+0.00}$ |
| data_entry | 0.00 | 0.11 | 0.00 | $0.07_{+0.07}$ | $0.10_{-0.01}$ | $0.00_{+0.00}$ |
| game_playing | 0.00 | – | – | $0.00_{+0.00}$ | – | – |
| information_retrieval | 0.14 | 0.00 | 0.00 | $0.43_{+0.29}$ | $0.11_{+0.11}$ | $0.00_{+0.00}$ |
| math_counting | 0.00 | 0.00 | 0.00 | $0.00_{+0.00}$ | $0.33_{+0.33}$ | $0.00_{+0.00}$ |
| memorization | 0.00 | 0.00 | 0.00 | $0.00_{+0.00}$ | $1.00_{+1.00}$ | $0.00_{+0.00}$ |
| multi_app | 0.00 | 0.00 | 0.00 | $0.00_{+0.00}$ | $1.00_{+1.00}$ | $0.00_{+0.00}$ |
| parameterized | 0.09 | 0.09 | 0.06 | $0.41_{+0.32}$ | $0.18_{+0.09}$ | $0.11_{+0.05}$ |
| repetition | 0.00 | 0.00 | 0.20 | $0.50_{+0.50}$ | $0.00_{+0.00}$ | $0.20_{+0.00}$ |
| requires_setup | 0.00 | 0.00 | 0.00 | $0.67_{+0.67}$ | $0.00_{+0.00}$ | $0.00_{+0.00}$ |
| screen_reading | 0.08 | 0.00 | 0.11 | $0.50_{+0.42}$ | $0.33_{+0.33}$ | $0.11_{+0.00}$ |
| search | 0.00 | 0.00 | 0.00 | $0.73_{+0.73}$ | $0.20_{+0.20}$ | $0.00_{+0.00}$ |
| transcription | 0.00 | 0.00 | 0.00 | $0.00_{+0.00}$ | $0.50_{+0.50}$ | $0.00_{+0.00}$ |
| untagged | 0.40 | 0.00 | – | $0.80_{+0.40}$ | $0.00_{+0.00}$ | – |
| verification | 0.86 | – | – | $1.00_{+0.14}$ | – | – |

Table 10: Mean AndroidWorld success rate comparison between Qwen2.5VL-7B and UI-S1-7B across tags and difficulty levels, with improvement indicated (positive, negative, no change).

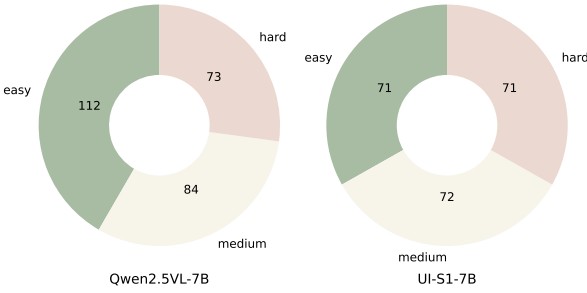

Figure 11: AndroidWorld task error count distribution grouped by difficulty.

We also display the error distribution. From the **difficulty** perspective (Figure 11), the most substantial improvement was observed in 'easy' tasks, where UI-S1-7B achieved a remarkable reduction of 41 errors compared to the base model. Following this, a moderate but significant performance gain was noted for 'medium' difficulty tasks. In stark contrast, the model's advantage diminished considerably for 'hard' tasks, showing only a marginal improvement with a reduction of two errors.

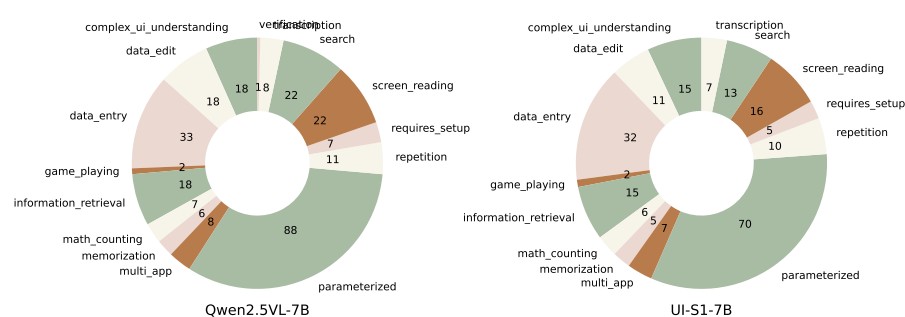

Figure 12: AndroidWorld task error count distribution grouped by task tag.

From a task **classification** perspective (Figure 12), while the proportional distribution of errors across different task categories remained largely consistent between the two models, UI-S1-7B demonstrated marked advancements in several key functional areas, such as `scree_reading`, `search`, `transcription`, `data_edit`, and `parameterized`.

## G.2 CASE COMPARISON

We showcase a complex cross-application task requiring information retention across multiple steps: creating a file in Markor with transaction details from an image viewed in Simple Gallery (as illustrated in Figure 13). The base model and Offline RL model exhibit action-thought inconsistency. For example, offline RL terminate prematurely after planning to navigate to the next app, likely due to overfitting to local rewards without considering future objectives. The SFT model loses critical information and executes redundant actions like attempting to create a file that already exists. In contrast, our model successfully records the critical information throughout the 12-step sequence, correctly recording "2023-03-23, Monitor Stand, $33.22" in CSV format. This demonstrates its effectiveness in learning robust multi-turn behaviors with consistent reasoning-action alignment. Additional case studies are provided in Appendix G.3 and failure analysis in Appendix G.4.

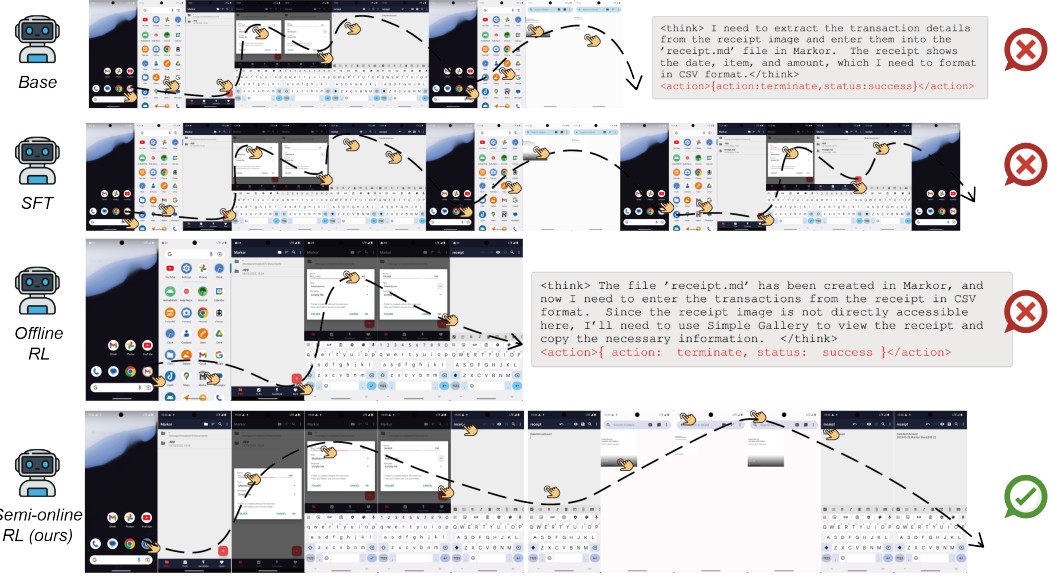

Figure 13: A cross-app and memorable task case in AndroidWorld. The instruction is *"Create a file in Markor, called receipt.md with the transactions from the receipt.png. Use Simple Gallery to view the receipt. Please enter transactions in csv format including the header 'Date, Item, Amount'."*

## G.3 MORE CASES

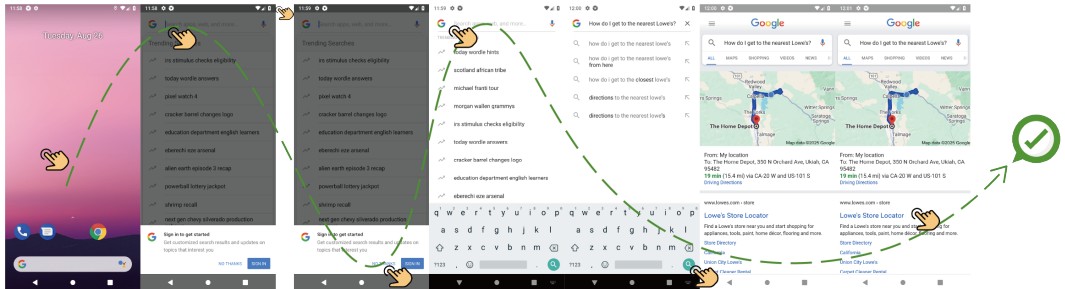

Figure 14: A successful task case encountering *sign in notes* in **AITW-Gen**. The instruction is *"How do I get to the nearest Lowe's?"*.

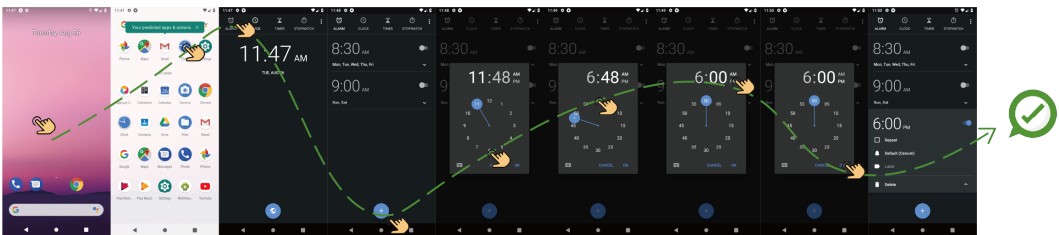

Figure 15: A successful task case in **AITW-Gen**. The instruction is *"Set an alarm for 6pm"*.

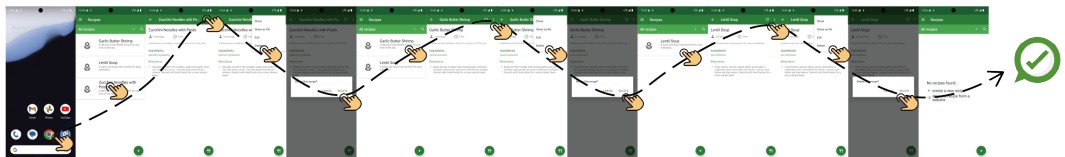

Figure 16: A successful task case in **AndroidWorld**. The instruction is *"Delete the following recipes from Broccoli app: Zucchini Noodles with Pesto, Garlic Butter Shrimp, Lentil Soup."*

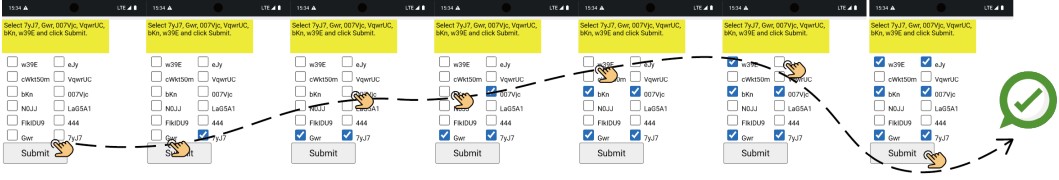

Figure 17: A successful task case in **MiniWob++**. The instruction is *"Follow the instructions shown on the top of the screen: Select 7yJ7, Gwr, 007Vjc, VqwrUC, bKn, w39E and click Submit."*

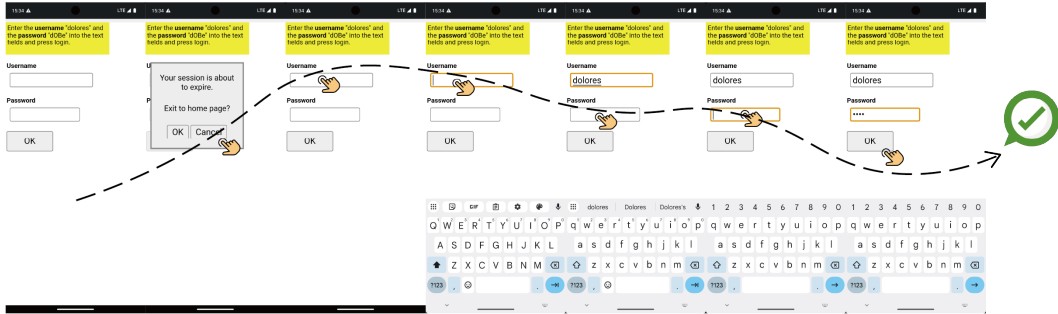

Figure 18: A successful task case in **MiniWob++**: *"Follow the instructions shown on the top of the screen: Enter the username dolores and the password dOBe into the text fields and press login."*.

## G.4 BAD CASE

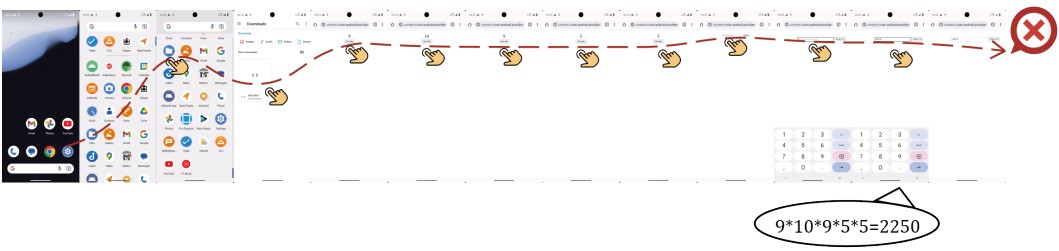

Figure 19: A failed task case in **AndroidWorld**. The instruction is *"Open the file task.html in Downloads in the file manager; when prompted open it with Chrome. Then click the button 5 times, remember the numbers displayed, and enter their product in the form."*.

## H  PROMPT FOR TRAINING AND INFERENCE

**System prompt:**
You are a GUI agent. You are given a task and your action history, with screenshots. You need to perform the next action to complete the task.
**Output Format**

```
<think> ... </think>
<action> ... </action>
```

**Action Space**
You can perform the following actions:
- key: Perform a key event on the mobile device using adb's 'keyevent' syntax.
- click: Click the point on the screen with specified (x, y) coordinates.
- long_press: Press the point on the screen with specified (x, y) coordinates for a specified number of seconds.
- swipe: Swipe from starting point with specified (x, y) coordinates to endpoint with specified (x2, y2) coordinates.
- type: Input the specified text into the activated input box.
- answer: Output the specified answer.
- system_button: Press the specified system button: Back, Home, Menu, or Enter.
- open: Open an application on the device specified by text.
- wait: Wait for a specified number of seconds for changes to occur.
- terminate: Terminate the current task and report its completion status: success or failure.
The arguments you can use are:
- coordinate: (x, y): The x and y pixels coordinates from the left and top edges.
- coordinate2: (x, y): The x and y pixels coordinates from the left and top edges for the endpoint of a swipe.
- text: Text input required by actions like 'key', 'type', 'answer', and 'open'.
- time: The time in seconds required by actions like 'long_press' and 'wait'.
- button: System buttons available for pressing: Back, Home, or Enter. Possible values: Back, Home, Menu, Enter.
- status: The completion status of a terminated task. Possible values: success, failure.
Format your output as a JSON object with the selected action and its arguments at the same level.
**Example Output**

```
<think>...</think>
<action>{"action": "key", "text": "<value>"}
```

**Note**
- Planing the task and explain your reasoning step-by-step in 'think' part.
- Write your action in the 'action' part according to the action space.
- If the query asks a question, please answer the question through the answer action before terminating the process.
- Swipe the screen to find the File Manager app if needed.
**User prompt:**
User Instruction: USER INSTRUCTION
**Assistant prompt:**
HISTORY RESPONSES
HISTORY IMAGES

# I  PROMPT FOR THOUGHT GENERATION

**System prompt:**
End-to-End Model Thought Integration
**Integration Requirements**

- Write the thought process from a global goal, the action history, thought history and screenshot history.
- The reasoning logic must satisfy:
  - Begin by reviewing the global task objective.
  - Inherit the context and decisions from historical steps.
  - Incorporate the manager's planning logic.
  - Derive actions that fully align with the operator's output.

**Output Format**

```
<think>
[A coherent reasoning process, reflecting task decomposition,
environmental observation, and iterative decision-making]
</think>
```

**Output Example**

```
<think>
The current task requires checking the order status of
DeepSeek. Access to the official website and locating the login
entry have been completed. Based on the page loading result,
the login form is ready. Authentication information needs
to be filled: the username has already been entered as
"DeepSeek," and now the password must be entered.
</think>
```

**Key Design Notes**

- Explicitly require the global task objective to ensure the end-to-end model always anchors to the core goal.
- Enforce structured historical records to prevent information loss.
- Logic consistency mechanism.
- The thought process should naturally connect historical conclusions with the current manager's planning.
- Transform the manager's planning into autonomous decisions phrased as "According to the requirements, determine..."
- Translate operator actions into imperative statements phrased as "Execute..."
- Do not mention any coordinates in <think> ... </think>.

**Global Task Objective**
USER INSTRUCTION
- If this isn't the target app for your operation, you can use open operation to navigate to the correct application.
- You can use Next Action Hint to guide the think process, but within the think section, you must conceal the fact that hints were received.
- Please integration the thought of current manager and operation into <think> ... </think> in English.
**Assistant prompt:**
HISTORY RESPONSES
HISTORY IMAGES

## J PROMPT FOR GPT-4O TO EVALUATE MINIWOB++ TASK

**System prompt:**
You're an expert in evaluating whether the Screenshot successfully completes the Task.
================================Examples================================
**Task**: Open the settings. Q: What should I expect to see on the screenshot if I've opened the settings? A: I should expect to see I'm in the settings app. The screenshot shows the home screen of a mobile device, with various app icons displayed, including the settings app icon, but the settings app is not opened.
Status: failure Screenshot: SCREENSHOT
**Task**: Find hotels in Washington DC Q: What should I expect to see on the screenshot if I've searched for hotels in Washington, DC? A: I should expect to see I'm in a search results page for hotels in Washington, DC. The screenshot shows a Google search page with the search field populated with the query "hotels in washington dc" and a list of suggested searches related to hotels in Washington, DC, but it does not show any search results for hotels in Washington, DC.
Status: failure Screenshot: SCREENSHOT
**Task**: What's a good restaurant in Portland? Q: What should I expect to see on the screenshot if I've searched for a good restaurant in Portland? A: I should expect to see I'm in a search results page for a good restaurant in Portland. The screenshot shows a Google search page with a search input field for "good restaurant in portland" and a map results preview showing business locations near Portland, like "Li Pigeon", "Portland City Grill", and "Higgins".
Status: success Screenshot: SCREENSHOT
**Task**: What's on the menu at In-N-Out? Q: What should I expect to see on the screenshot if I've searched for the menu at In-N-Out? A: I should expect to see a menu page for In-N-Out, including product names, thumbnails and prices. The screenshot shows a Google search page with a search input field for "In-N-Out menu" and some page snippets of In-N-Out indicating potential menu items, but does not actually show the actual menu.
Status: failure Screenshot: SCREENSHOT
**Task**: What's the news in Suriname? Q: What should I expect to see on the screenshot if I've searched for the news in Suriname? A: I should expect to see some news in Suriname, such as someone did something or some accident happens in Suriname. The screenshot shows a Google search page with a search input field for "Suriname news today" and some page snippets indicating potential news items, but does not actually show the news.
Status: failure Screenshot: SCREENSHOT
**Task**: What's the weather like in Chicago? Q: What should I expect to see on the screenshot if I've searched for the weather in Chicago? A: I should expect to see some exact values like temperature, humidity, wind speed, and weather condition in Chicago. The screenshot shows a Google search page with a search input field for "weather in Chicago" and some page snippets indicating potential weather information. Although one page snippet contains some weather information, the information is not comprehensive enough to determine the weather in Chicago.
Status: failure Screenshot: SCREENSHOT
**Task**: Set an alarm for 6pm. Q: What should I expect to see on the screenshot if I've set an alarm for 6pm? A: I should expect to see some alarms including a 6pm alarm activated in the clock app. The screenshot shows an attempt to set an alarm for 6pm in the clock app, but the alarm is not set yet.
Status: failure Screenshot: SCREENSHOT

## K THE USE OF LLM

In the preparation of this manuscript, we utilized large language models (LLMs) as a writing aid. Their use was strictly limited to grammar checking, syntax refinement, and language polishing to improve the clarity and readability of the text. We affirm that all intellectual contributions, including the core ideas, methodology, data analysis, and the substantive writing, are the original work of the authors. The authors retain full responsibility for the content and conclusions of this paper.

