# OpenReview forum: "UI-S1: Advancing GUI Automation via Semi-online Reinforcement Learning"
_ICLR.cc/2026/Conference — ICLR 2026 Conference Withdrawn Submission_

### Official Review · Reviewer_1YPk · 2025-10-21

**Soundness:** 1
**Presentation:** 1
**Contribution:** 2
**Rating:** 2
**Confidence:** 3

**Summary:**

The manuscript deals with reinforcement learning for GUI development. An approach called semi-online RL is introduced.

**Strengths:**

* The use of LLMs to support RL is an interesting research topic.

**Weaknesses:**

* The presentation is unclear.
* Terms are used that have not been defined.
* The integration into existing RL literature is insufficient. In particular, model-based RL and rollouts in model-based RL are not discussed.
* The entire method remains unclear.
* In my opinion, it is completely unacceptable to claim the term “semi-online RL” without going into detail about existing RL techniques.
* No justification is given as to why the results are considered statistically significant.

Details and further comments:

The sentence “offline RL [...] struggles with multi-step task execution for lack of trajectory-level reward signals; online RL captures these signals through environment interaction, but suffers from sparse rewards” does not seem to make sense to me.

The sentence “Semi-online RL introduces discounted future returns into the reward computation” does not seem to make sense to me.

The terms policy, policy model, and agent are used without it being clear what differences are meant by them, or whether they are just different terms for the same thing.

**Questions:**

Is it intended that the presented method will also bring advantages in classic RL benchmarks, such as Halfcheetah?

---

### Official Review · Reviewer_rNwR · 2025-10-22

**Soundness:** 3
**Presentation:** 3
**Contribution:** 2
**Rating:** 4
**Confidence:** 5

**Summary:**

This paper introduces **UI-S1**, a framework that bridges the gap between offline and online reinforcement learning for GUI automation. Traditional offline RL is stable but lacks multi-turn reasoning, while online RL captures long-horizon dependencies but requires costly real-time interactions. UI-S1 proposes a **Semi-Online RL** approach that simulates online dynamics entirely on **offline expert trajectories**, achieving both data efficiency and contextual consistency.

UI-S1’s core innovation is the **Patch Module**, which “repairs” rollouts when the model’s predicted actions deviate from expert trajectories—allowing continued training without terminating early. Combined with a **dual-level policy optimization** (step-level for immediate accuracy, episode-level for overall success), this enables long-horizon learning without environment interaction. The authors also introduce the SOP metric, an offline measure that strongly correlates (R² = 0.93) with real-world task success, providing a reliable proxy for evaluating GUI agents.

**Strengths:**

The paper introduces a Semi-Online Reinforcement Learning framework that bridges offline and online RL without requiring live environment interaction. This hybrid formulation is an original contribution that removes major practical barriers of online GUI agents while retaining long-horizon reasoning capabilities. The approach substantially improves the scalability of GUI automation models by eliminating the need for interactive training. This has high significance for deploying intelligent agents in real-world software automation.

**Weaknesses:**

1. The so-called Semi-Online approach does not actually utilize any data from real online interactions. It remains fundamentally an offline training method that relies on pre-collected expert trajectories, with more fine-grained optimization at the step level. Therefore, it would be more convincing if the paper compared UI-S1 directly against other expert-data-based training methods on the same datasets, such as SFT and the following related works:
   - *DigiRL: Training In-The-Wild Device-Control Agents with Autonomous Reinforcement Learning* (https://arxiv.org/abs/2406.11896)
   - *UI-R1: Enhancing Efficient Action Prediction of GUI Agents by Reinforcement Learning* (https://arxiv.org/abs/2503.21620)

2. The paper’s key contribution—Semi-Online training—heavily depends on expert trajectories for the Patch Module, yet the source and composition of these expert data are not described in sufficient detail.
   Specific questions remain unanswered:
   - How were the expert trajectories collected or constructed?
   - What is the proportion of data from different sources within the reported 2,000 samples?
   - If these trajectories are verified as “perfect rollouts,” were they also used in the SFT stage? If so, what are the results of training solely with these verified trajectories for SFT?

3. Section 3.4 assigns different weights to various reward components, but the paper does not provide a theoretical or empirical justification for these specific weights.
   Moreover, the SOP method (Semi-Online Performance) is not explicitly defined—its mathematical formulation is unclear, and it is ambiguous whether SOP represents the same quantity as the weighted reward function described in Section 3.4.

4. The results show that Off-Policy Thought Patch consistently underperforms both Thought-Free and On-Policy Patch methods. However, the paper does not explain the positive motivation or utility of this variant—its conceptual necessity and practical benefit remain unclear.

**Questions:**

The questions are already included within the weaknesses section.

---

### Official Review · Reviewer_e69T · 2025-10-23

**Soundness:** 2
**Presentation:** 3
**Contribution:** 2
**Rating:** 4
**Confidence:** 4

**Summary:**

The paper introduces Semi-online Reinforcement Learning (RL) for GUI agents, a framework that trains on offline trajectories while simulating online rollouts through a Patch Module. The method optimizes performance using dual-level advantages combined with discounted returns, achieving strong results on AndroidWorld, AITW, and MiniWob++ benchmarks.
Additionally, the paper proposes the SOP metric, designed to effectively measure performance in semi-online settings.

**Strengths:**

- The paper is very well written and organized.
- The experiments demonstrate the effectiveness of Semi-Online RL. Across benchmarks like AndroidWorld, AITW-Gen, and MiniWob++, the proposed UI-S1 model consistently outperforms baselines such as SFT and Offline RL.
- The paper provides thorough ablation studies covering discount factors, training paradigm combinations (SFT + RL), the role of episode-level advantages, and patch threshold sensitivity.

**Weaknesses:**

- Reliance on expert trajectories and "oracle" dynamics. The rollout process depends on expert next states when the actions align, and on patched expert actions when they don’t. This introduces an negative bias in return estimation and restricts the model’s exposure to true on-policy states, potentially limiting robustness and generalization.

- Limited methodological novelty. Compared with offline GRPO or "step-level GRPO", the main improvements of this work lie in the advantage computation and the organization of history. These changes do not represent substantial algorithmic innovation.

**Questions:**

- Insufficient analysis of offline GRPO improvements. It is especially important to analyze why offline GRPO underperforms SFT in multi-turn tasks but outperforms in single-turn tasks, and how the authors’ modifications specifically address the shortcomings of offline GRPO in multi-turn interactions.

- Unclear rationale behind the success of the Thought-Free Patch. The results show that the Thought-Free Patch performs comparably to, or even better than the On-Policy Thought Patch. However, since the Thought-Free Patch operates entirely out of distribution, this outcome seems counterintuitive and requires further explanation or theoretical justification.

- Lack of clarity on the effect of $\epsilon$. A larger $\epsilon$ corresponds to more training data, and according to the definition of advantage, $\epsilon$ should not affect advantage estimation. Therefore, it is unclear why increasing $\epsilon$ leads to worse performance, this phenomenon needs more analysis and discussion.

**Details Of Ethics Concerns:**

No ethics concerns

---

### Official Review · Reviewer_6v2V · 2025-10-31

**Soundness:** 2
**Presentation:** 3
**Contribution:** 3
**Rating:** 4
**Confidence:** 3

**Summary:**

The paper presents UI-S1, a framework for GUI automation using a method termed Semi-Online Reinforcement Learning that bridges the gap between single-turn offline RL and multi-turn online RL.
It simulates online interaction using offline data through a Patch Module that corrects action mismatches and maintains training continuity.
A dual-level reward system captures both step-wise and long-term advantages, improving multi-step reasoning.
The authors also propose a new evaluation metric, Semi-Online Performance (SOP), which correlates strongly with real-world performance.
UI-S1-7B achieves state-of-the-art results among 7B models, outperforming prior methods on multiple GUI benchmarks.

**Strengths:**

- Bridges offline single-turn and online multi-turn RL: Introduces "Semi-Online RL", effectively combining the merits of both worlds.
- Efficient learning from static data: The Patch Module enables continued training despite action mismatches, improving data utilization and multi-turn learning.
- More Reliable evaluation metric: Proposes Semi-Online Performance (SOP), a fast and accurate proxy for real-world online performance (R² = 0.934).
- State-of-the-art results in multi-turn performance: The UI-S1-7B model achieves top performance across multiple GUI automation benchmarks among models of the same size

**Weaknesses:**

I believe the paper is proposing a reasonably novel method, shows good performance on benchmarks, and overall has a demonstrated merit over prior works, however I am utterly confused by the terminology. To highlight just the most extreme cases:

- "Traditional offline RL trains on static trajectories where each step conditions on expert demonstrations"
- Similarly: "Traditional offline RL optimizes only for immediate step-wise accuracy, resulting in multi-turn planning failure. "

--> To me, these statements appear to be inaccurate or, at minimum, a rebranding of well-established terminology. There are hundreds of papers on offline reinforcement learning that explicitly address long-horizon (i.e., multi-turn) decision-making by maximizing discounted cumulative reward—arguably the standard definition of RL. The setting you describe (offline + expert data + single-step optimization at training time) seems to correspond closely to what is typically referred to as behavior cloning (e.g., [1–3]). Could you please clarify what makes this rebranding necessary / what I am missing here?

[1] Foster, Dylan J., Adam Block, and Dipendra Misra. "Is behavior cloning all you need? understanding horizon in imitation learning." Advances in Neural Information Processing Systems 37 (2024): 120602-120666.

[2] Kumar, Aviral, et al. "Should i run offline reinforcement learning or behavioral cloning?." International conference on learning representations. 2021.

[3] Wang, Zhendong, Jonathan J. Hunt, and Mingyuan Zhou. "Diffusion policies as an expressive policy class for offline reinforcement learning." arXiv preprint arXiv:2208.06193 (2022).

In addition, the term semi-online RL seems potentially misleading, as—if I understand correctly—no online interactions are involved in your setup. Overall, it appears that the manuscript does not sufficiently engage with a substantial body of prior work in this area.

Finally, I read some of the claims as a bit overstated, e.g. "[...] Semi-Online RL successfully bridges both capabilities rather than trading one for the other", "[...] validating that Semi-online RL doesn’t sacrifice single-turn capabilities.", etc.
When I compare single-turn performance of the proposed model with e.g. UI-TARS-7B, averaging over the 8 columns I get: 67.5 vs 79.5, which appears a quite significant drop compared to this model.

I'm happy to raise the score if the above can be addressed.

**Questions:**

see weaknesses

---

### Note · Authors · 2025-11-13

I have read and agree with the venue's withdrawal policy on behalf of myself and my co-authors.